# InfMasking: Unleashing Synergistic Information by Contrastive Multimodal Interactions

**Liangjian Wen**[1,2,8*], **Qun Dai**[1*], **Jianzhuang Liu**[3], **Jiangtao Zheng**[1], **Yong Dai**[4‡], **Dongkai Wang**[1†], **Zhao Kang**[5], **Jun Wang**[1], **Zenglin Xu**[6,7], **Jiang Duan**[1†]

[1]School of Computing and Artificial Intelligence, Southwestern University of Finance and Economics
[2]Engineering Research Center of Intelligent Finance, Ministry of Education,Southwestern University of Finance and Economics
[3]Shenzhen Institutes of Advanced Technology, Chinese Academy of Sciences
[4]X-Humanoid
[5]University of Electronic Science and Technology of China
[6]Shanghai Academy of AI for Science
[7]Artificial Intelligence Innovation and Incubation Institute, Fudan University
[8]Artificial Intelligence and Digital Finance Key Laboratory of Sichuan Province
{wlj6816,daiqun1124}@gmail.com

## Abstract

In multimodal representation learning, synergistic interactions between modalities not only provide complementary information but also create unique outcomes through specific interaction patterns that no single modality could achieve alone. Existing methods may struggle to effectively capture the full spectrum of synergistic information, leading to suboptimal performance in tasks where such interactions are critical. This is particularly problematic because synergistic information constitutes the fundamental value proposition of multimodal representation. To address this challenge, we introduce InfMasking, a contrastive synergistic information extraction method designed to enhance synergistic information through an **Inf**inite **Masking** strategy. InfMasking stochastically occludes most features from each modality during fusion, preserving only partial information to create representations with varied synergistic patterns. Unmasked fused representations are then aligned with masked ones through mutual information maximization to encode comprehensive synergistic information. This infinite masking strategy enables capturing richer interactions by exposing the model to diverse partial modality combinations during training. As computing mutual information estimates with infinite masking is computationally prohibitive, we derive an InfMasking loss to approximate this calculation. Through controlled experiments, we demonstrate that InfMasking effectively enhances synergistic information between modalities. In evaluations on large-scale real-world datasets, InfMasking achieves state-of-the-art performance across seven benchmarks. Code is released at https://github.com/brightest66/InfMasking.

---

[*]Equal contribution.
[†]Corresponding author.
[‡]Project leader.

39th Conference on Neural Information Processing Systems (NeurIPS 2025).

# 1   Introduction

Multimodal contrastive learning has revolutionized representation learning by enabling the integration of diverse data modalities, such as text, images, and audio, into a unified latent space. This paradigm leverages contrastive loss [37, 9] to align features from different modalities as pioneered by foundational works like CLIP [38] and ALIGN [23] in vision-language tasks. These models demonstrate the power of aligning multimodal features to capture shared patterns across data sources, enabling versatile downstream applications. However, current approaches often rely on the restrictive *multiview redundancy assumption* [40, 43, 46], which posits that a modality is (approximately) sufficient for the prediction of downstream tasks and contains the same task-relevant information. This assumption derives from multi-view learning and is limited in real-world multimodal settings because many multimodal tasks involve minimal shared information.

The shortcomings of this redundancy-centric perspective become increasingly apparent when we examine the multifaceted nature of multimodal interactions. As illustrated in [6, 12], these interactions can be classified into three fundamental categories: redundancy, uniqueness, and synergy. Redundancy refers to scenarios where a modality can independently perform the task due to overlapping, shared information. Uniqueness describes cases where only one modality possesses all the requisite information for task completion. Synergy, arguably the most significant yet elusive of the three, occurs when modalities provide complementary information that must be combined to achieve the desired outcome. These interaction types are not static; their predominance varies depending on the specific task, adding a layer of complexity to multimodal learning. For instance, a task might rely heavily on redundant information in one context, while another demands the synergistic integration of modalities to succeed. A compelling example of this is hateful meme detection [26], where synergy emerges when seemingly neutral modalities (such as an innocuous image and benign text) combine to create harmful content that neither conveys on its own. This highlights the critical importance of synergistic integration, as models must fuse information from different modalities cues to uncover implicit biases or offensive implications that are only apparent in their joint context, enabling more effective identification and mitigation of such content in real-world applications. Consequently, task-agnostic multimodal representations must necessarily encompass the full spectrum of multimodal interactions that transcend mere informational redundancy.

Recently, FactorCL [29] explicitly decomposes shared and unique representations, enabling the estimation of redundancy and unique interactions beyond multi-view redundancy. However, its factorized approximation of multimodal interactions is prone to cumulative errors and fails to capture synergistic information effectively. In contrast, [12] integrates features from all modalities to derive a common representation and subsequently maximizes the mutual information between this representation and its augmented variants, as well as between the common representation and its corresponding unimodal features. Although this approach facilitates the capture of redundant, unique, and synergistic information across modalities, it primarily emphasizes enhanced redundant and unique interactions. Synergistic information is captured by maximizing the mutual information between the common representation and its augmented variants. Such handling may prove inadequate for tasks that heavily rely on complex inter-modal synergy. The complexity of synergistic interactions lies not merely in modalities providing complementary information but in how these modalities combine through specific interaction patterns to produce outcomes unattainable by any single modality alone. Such interactions may involve nonlinear dependencies or context-dependent dynamics. Hence, effectively capturing the full spectrum of synergistic information remains a significant challenge.

To address this challenge, we introduce a contrastive synergistic information extraction method via infinite masking (denoted as InfMasking). Specifically, InfMasking stochastically occludes a substantial proportion of features from each modality during the fusion process. This masking preserves only partial information, creating fused representations with varied synergistic patterns. Subsequently, unmasked fused representations are aligned with these masked ones via mutual information maximization to encode comprehensive synergistic information. The infinite masking strategy enables InfMasking to capture richer synergistic interactions by exposing the model to diverse combinations of partial modality information during training. However, the computation of mutual information estimates with infinite masking is computationally prohibitive. To address this issue, we derive an InfMasking loss to approximate the calculation of this loss function. Empirically, InfMasking effectively captures synergistic information between modalities in controlled environments with known interaction types. When tested on real-world datasets across diverse domains (healthcare,

robotics) and data types (image, text, audio), InfMasking achieves state-of-the-art results on seven multimodal tasks involving two or three modalities.

## 2 Preliminary: Contrastive Multimodal Interactions

Consider $X_1, X_2, \ldots, X_n$ as random variables, each representing a distinct modal data (e.g., image, text, audio, or tabular data), alongside a downstream task $Y$. The objective is to derive an effective multimodal latent variable $Z_\theta = f_\theta(X)$, where $X = (X_1, \ldots, X_n)$ and $\theta$ detnotes the parameter of multimodal encoder. Multimodal interactions refer to the process of extracting and integrating information from multiple data sources, such as text, image, audio, or tabular data, to form a cohesive representation for downstream tasks. These interactions can be categorized into three types: redundancy (R), where information is shared across modalities; uniqueness (U), where information is specific to a single modality; and synergy (S), where complementary information emerges only when modalities are combined.

To understand multimodal interactions, we can leverage Partial Information Decomposition (PID) [49, 6], a theoretical framework that dissects the mutual information between multiple variables. For analytical tractability, we focus on the case of $n = 2$. Consider two modalities $X_1$ and $X_2$ and a task $Y$. The mutual information $I(X_1, X_2; Y)$ quantifies the total task-relevant information provided by both modalities jointly. According to PID, this can be decomposed as: $I(X_1, X_2; Y) = R + S + U_1 + U_2$, where $R$ represents redundant information, common to both $X_1$ and $X_2$, $S$ represents synergistic information, emerging only from the combination of $X_1$ and $X_2$, and $U_1$ and $U_2$ represent unique information specific to $X_1$ and $X_2$, respectively. This decomposition is supported by consistency equations derived from the chain rule of mutual information:

$$I(X_1; Y) = R + U_1, \quad I(X_2; Y) = R + U_2, \quad I(X_1; X_2; Y) = R - S.$$

Effectively capturing these interactions is fundamental to multimodal learning, as they collectively enhance a model's ability to generalize across diverse tasks. Therefore, an effective multimodal representation $Z_\theta$ must capture information relevant to a downstream task $Y$, preserving the mutual information such that $I(Z; Y) = I(X; Y)$. In self-supervised learning, however, $Y$ remains unspecified during the representation learning phase, presenting a unique challenge. To address this, [12] extends the contrastive learning principle of multiview redundancy to multimodal contexts:

**Assumption 1 (*Minimal label-preserving multimodal augmentations*)** *A set $\mathcal{T}^*$ of multimodal transformations exists, such that for any $t \in \mathcal{T}^*$ and $X' = t(X)$, the mutual information $I(X; X') = I(X; Y)$ holds, preserving the information with label $Y$.*

Assumption 1 is justified within the framework of multimodal representation learning. It enables a broader range of augmentations, as it is not limited to the set $\mathcal{T}_c^\star = \{t(X) = (t_1(X_1), t_2(X_2)), \ldots, t_n(X_n)\}$. We define $Z'_\theta = f_\theta(X')$, where $X' = t(X)$ with $t \in \mathcal{T}$ representing multimodal augmentation. Considering the data processing inequalities for the Markov chains $X \to X' \to Z'_\theta$ and $Z'_\theta \to X \to Z_\theta$, we can establish the following mutual information bounds: $I(Z_\theta; Z'_\theta) \leq I(X; Z'_\theta) \leq I(X; X')$. According to these inequalities and Assumption 1, we can prove the following lemmas:

**Lemma 1** *When optimizing the function $f_\theta$ to maximize mutual information $I(Z_\theta; Z'_\theta)$, and under the assumption that the network $f_\theta$ possesses sufficient expressivity, we observe that in the optimal parameter configuration: $I(Z_{\theta^\star}, Z'_{\theta^\star}) = I(X, X') = I(X, Y)$.*

For $Z_\theta$ to serve as an effective representation of $X$, it must adequately preserve and encode all task-relevant information inherent in $X$. We note that $I(X; Y) = I(X_1, X_2; Y) = R + S + U_1 + U_2$. Based on Lemma 1, we can learn common multimodal representations $Z_\theta$ and quantify all multimodal interactions beyond redundancy by maximizing the mutual information $I(Z_\theta; Z'_\theta)$.

**Lemma 2** *Suppose $f_{\theta^\star}$ is optimal, meaning it maximizes $I(Z_{\theta^\star}, Z'_{\theta^\star})$. Then, the equality $I(Z_{\theta^\star}, Y) = I(X', Y)$ holds. For the special case where $T = \{t_i\}$ such that $X' = t_i(X) = X_i$ and $Z'_{\theta^\star} = f_{\theta^\star}(X) = Z_i$ for $i \in \{1, 2\}$, the following holds: $I(Z_i; Y) = I(X_i; Y) = R + U_i$.*

For unimodal representations $Z_i$ where $i \in \{1, 2\}$ to effectively represent $X_i$, each representation must encode the information $I(X_i; Y) = R + U_i$ contained in the corresponding modality. According to Lemma 2, we can learn optimal unimodal representations $Z_i$ and quantify both redundant and unique multimodal interactions by maximizing the mutual information $I(Z_i; Z'_\theta)$ and $I(Z_i; Z_\theta)$.

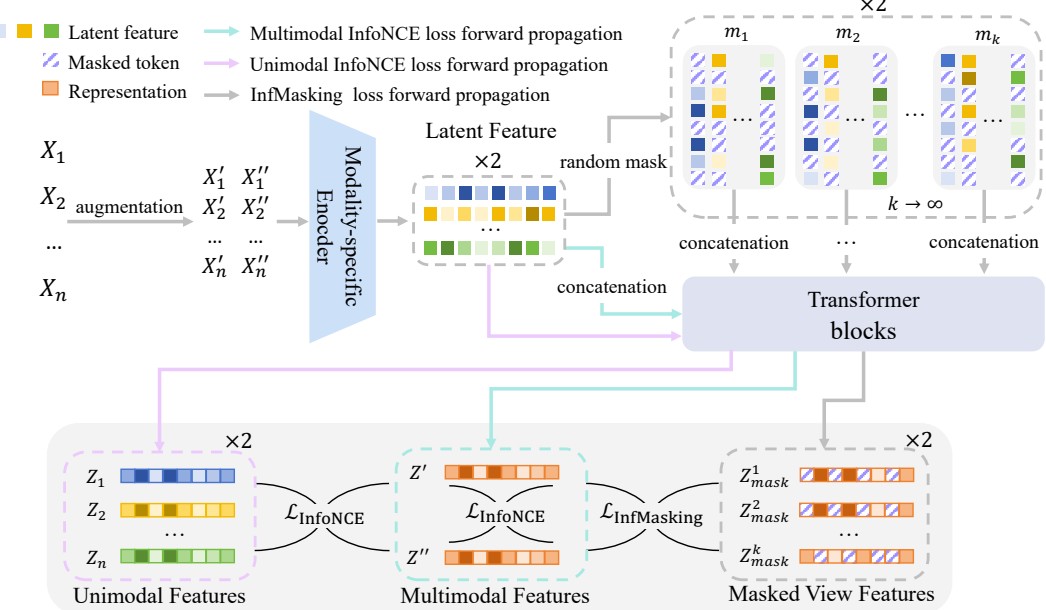

Figure 1: The overall pipeline of InfMasking. Given $n$ modalities $X = (X_1, X_2, \ldots, X_n)$, we augment them to obtain $X'$ and $X''$, which are then encoded independently by modality-specific encoders to extract latent features. These features are processed in three ways: (1) All modality features are concatenated and input into a Transformer block, yielding fused features $Z'$ and $Z''$; (2) Each modality feature is individually input into a Transformer block, producing unimodal features $Z_1, Z_2, \ldots, Z_n$; (3) Features of each modality are randomly masked, concatenated, and input into a Transformer block, repeated $k$ times to obtain $Z^1_{\text{mask}}, Z^2_{\text{mask}}, \ldots, Z^k_{\text{mask}}$.

## 3 Unleashing Synergistic Information through Infinite Masking

### 3.1 The General Framework

The proposed framework, termed InfMasking, is a multimodal contrastive interaction method designed to enhance synergistic information across modalities by leveraging infinite masked views. The overall pipeline of InfMasking is illustrated in Fig. 1 and consists of two primary stages: modality-specific latent feature encoding and multimodal feature fusion via a Transformer. Given an input set of $n$ modalities $X = (X_1, X_2, \ldots, X_n)$, we obtain $X'$ and $X''$ through an augmentation process. Subsequently, $X'$ and $X''$ are processed by modality-specific encoders, where each modality is encoded independently to extract latent features. As shown in Fig. 1, these modality features are then processed in parallel through three distinct ways: (1) All modality features are concatenated and input into a Transformer block, yielding fused features $Z'$ and $Z''$; (2) Each modality feature is individually input into a Transformer block, producing unimodal features $Z'_1, Z'_2, \ldots, Z'_n$ and $Z''_1, Z''_2, \ldots, Z''_n$; (3) Features of each modality are randomly masked, then concatenated and input into a Transformer block to obtain fused features. This process is executed $k$ times, resulting in $Z'^1_{\text{mask}}, Z'^2_{\text{mask}}, \ldots, Z'^k_{\text{mask}}$ and $Z''^1_{\text{mask}}, Z''^2_{\text{mask}}, \ldots, Z''^k_{\text{mask}}$.

Based on Lemma 1 and Lemma 2, [12] proposes a contrastive multimodal (CoMM) learning approach to learn task-agnostic multimodal representations by modeling multimodal interactions, including redundancy, uniqueness, and synergy. CoMM estimates the mutual information using the InfoNCE loss: $\hat{I}_{\text{NCE}}(Z, Z') = \mathbb{E}_{z, z'_{\text{pos}} \sim p(Z, Z')} \left[ \log \frac{\exp(z^T z'_{\text{pos}}/\tau)}{\exp(z^T z'_{\text{pos}}/\tau) + \sum_{z'_{\text{neg}}} \exp(z^T, z'_{\text{neg}}/\tau)} \right]$, where $\tau$ is a temperature parameter. Hence, its training objective is formulated as follows:

$$\mathcal{L}_{\text{CoMM}} = - \underbrace{\hat{I}_{\text{NCE}}(Z', Z'')}_{\approx R + S + \sum_{i=1}^n U_i} - \sum_{i=1}^n \tfrac{1}{2} \underbrace{\left( \hat{I}_{\text{NCE}}(Z_i, Z') + \hat{I}_{\text{NCE}}(Z_i, Z'') \right)}_{\approx R + U_i}. \tag{1}$$

While the first term theoretically quantifies redundancy, synergy, and the unique information across modalities, empirical evidence indicates that its practical performance exhibits notable limitations. Hence, the second term constitutes the fundamental component of CoMM, specifically designed to strengthen both unique and redundant interaction patterns. However, enhancing synergistic interactions remains a substantial challenge in this framework.

Synergy is a complex interaction that arises when different modalities provide complementary information, necessitating their integration for effective task performance. We aim to learn a multimodal representation that captures all three types of interactions, with a particular emphasis on enhancing synergistic information. To achieve this, we introduce InfMasking , a novel approach that leverages infinite masking to enhance the modeling of multimodal interactions. Our training objective is formulated as follows:

$$\mathcal{L}_{\text{Total loss}} = - \underbrace{\hat{I}_{\text{NCE}}(Z', Z'')}_{\approx R + S + \sum_{i=1}^{n} U_i} - \sum_{i=1}^{n} \frac{1}{2} \underbrace{\left( \hat{I}_{\text{NCE}}(Z_i, Z') + \hat{I}_{\text{NCE}}(Z_i, Z'') \right)}_{\approx R + U_i}$$
$$- \underbrace{\mathbb{E}_{\text{mask}} \left[ \hat{I}_{\text{NCE}}(Z'_{\text{mask}}, Z') + \hat{I}_{\text{NCE}}(Z''_{\text{mask}}, Z'') \right]}_{\mathcal{L}_{\text{InfMasking}}}, \tag{2}$$

where $\mathcal{L}_{\text{InfMasking}}$ represents our novel masking-based regularization term designed to specifically enhance synergistic interactions, as detailed in Section 3.2.

## 3.2 Contrastive Synergistic Information via Infinite Masking

In multimodal learning, capturing synergistic information—where different modalities provide complementary insights—is essential for tasks requiring integrated understanding. We propose a contrastive synergistic Information method via infinte masking to enhance synergistic interactions. It's core idea is to randomly mask a significant portion of the features from each modality during the fusion process. As shown in Fig. 1, we fuse all masked features from different modalities to obtain a fused representation $Z'_{mask}$ via a Transformer. Each time features from each modality are randomly masked, only partial information from each modality is retained. Consequently, after each masking operation, $Z'_{mask}$ contains distinct synergistic information. Then, by aligning $Z'_{mask}$ with its unmasked counterparts $Z'$ through maximizing their mutual information, $Z'$ are encouraged to capture distinct synergistic information. This process is repeated for $K$ times of masking, we can obtain the final training loss: $\frac{1}{K} \sum_{k=1}^{K} \hat{I}_{\text{NCE}}(Z'^{\,k}_{\text{mask}}, Z') + \hat{I}_{\text{NCE}}(Z''^{\,k}_{\text{mask}}, Z'')$. To enable the model to learn diverse forms of synergistic information, we allow $K$ to approach infinity through infinite masking, ultimately obtaining the masking loss $\mathcal{L}_{\text{InfMasking}}$ as follows:

$$\mathcal{L}_{\text{InfMasking}} = - \lim_{K \to \infty} \frac{1}{K} \sum_{k=1}^{K} \hat{I}_{\text{NCE}}(Z'^{\,k}_{\text{mask}}, Z') + \hat{I}_{\text{NCE}}(Z''^{\,k}_{\text{mask}}, Z'')$$
$$= -\mathbb{E}_{\text{mask}} \left[ \hat{I}_{\text{NCE}}(Z'_{\text{mask}}, Z') + \hat{I}_{\text{NCE}}(Z''_{\text{mask}}, Z'') \right]. \tag{3}$$

This infinite masking strategy enables InfMasking to capture richer synergistic interactions by exposing the model to diverse combinations of partial modality information during training. However, the estimation of $\mathbb{E}_{\text{mask}} \left[ \hat{I}_{\text{NCE}}(Z'_{\text{mask}}, Z') \right]$ and $\mathbb{E}_{\text{mask}} \left[ \hat{I}_{\text{NCE}}(Z''_{\text{mask}}, Z'') \right]$ is computationally expensive via random mask samples.

To address this issue, we derive a lower bound for $\mathbb{E}_{\text{mask}}\left[\hat{I}_{\text{NCE}}(Z'_{\text{mask}}, Z')\right]$ to optimize the InfMasking loss function Eq. (3). The detailed derivation is as follows:

$$\mathbb{E}_{\text{mask}}[\hat{I}_{\text{NCE}}(Z'_{\text{mask}}, Z')] = \mathbb{E}_{\text{mask}}[\mathbb{E}_{z' \sim p(Z')}\left[\log \frac{\exp(z'^T z'_{\text{mask}}/\tau)}{\exp(z'^T z'_{\text{mask}}/\tau) + \sum_{z'_{\text{neg}}} \exp(z'^T z'_{neg}/\tau)}\right]] \quad (4)$$

$$= \mathbb{E}_{z' \sim p(Z')}[\mathbb{E}_{\text{mask}}\left[(z'^T z'_{\text{mask}}/\tau) - \log[\exp(z'^T z'_{\text{mask}}/\tau) + \sum_{z'_{\text{neg}}} \exp(z'^T z'_{neg}/\tau)]\right]] \quad (5)$$

$$\geq \mathbb{E}_{z' \sim p(Z')}\left[z'^T \mathbb{E}_{\text{mask}}[z'_{\text{mask}}]/\tau - \log \mathbb{E}_{\text{mask}}[\exp(z'^T z'_{\text{mask}}/\tau) + \sum_{z'_{\text{neg}}} \exp(z'^T z'_{neg}/\tau)]\right] \quad (6)$$

The inequality Eq. (6) merges from the application of Jensen inequality on concave functions i.e., $\mathbb{E}_x \log(X) \leq \log \mathbb{E}_x[X]$. $z'_{\text{mask}}$ denotes the integrated representation derived from the fusion of all masked features across diverse modalities via the Transformer architecture.

Inspired by [7], we posit that $z'_{\text{mask}}$ follows a Gaussian distribution, formally expressed as $z'_{\text{mask}} \sim \mathcal{N}(\boldsymbol{\mu}_{z'_{\text{mask}}}, \boldsymbol{\Sigma}_{z'_{\text{mask}}})$, where $\boldsymbol{\mu}_{z'_{\text{mask}}}$ and $\boldsymbol{\Sigma}_{z'_{\text{mask}}}$ denote the mean vector and covariance matrix of $z'_{\text{mask}}$, respectively. This assumption is well-founded for two principal reasons. First, the masked embeddings tend to cluster around a central value in the embedding space, as they all inherently reflect aspects of the query's semantic nature. Second, the variance observed across feature dimensions can be interpreted as a representation of semantic differentiation in the ambient space, which aligns with established principles in distributional semantics. Under this assumption, we can derive:

**Lemma 3** *Let $\boldsymbol{\mu}_{z'_{\text{mask}}}$ and $\boldsymbol{\Sigma}_{z'_{\text{mask}}}$ be the Gaussian mean vector and covariance matrix of $z'_{\text{mask}}$, respectively. The lower bound of $\mathbb{E}_{\text{mask}}\left[\hat{I}_{NCE}(Z'_{\text{mask}}, Z')\right]$ can be obtained as follows:*
$\mathbb{E}_{\text{mask}}[\hat{I}_{NCE}(Z'_{\text{mask}}, Z')]$

$$\geq \mathbb{E}_{z' \sim p(Z')}\left[z'^T \boldsymbol{\mu}_{z'_{\text{mask}}}/\tau - \log[\exp(z'^T \boldsymbol{\mu}_{z'_{\text{mask}}}/\tau + \frac{z'^T \boldsymbol{\Sigma}_{z'_{\text{mask}}} z'}{2\tau^2}) + \sum_{z'_{\text{neg}}} \exp(z'^T z'_{neg}/\tau)]\right] \quad (7)$$

This allows us to approximate the mutual information between the masked and unmasked representations without requiring exhaustive sampling of all possible masks. A detailed proof is given in Appendix G.

## 4 Experiments

We perform experiments on both synthetic benchmarks and multiple large-scale real-world datasets to verify the effectiveness of InfMasking in learning representations from diverse modalities. To evaluate InfMasking's capacity to capture three essential aspects of multimodal interactions (*i.e.*, uniqueness, redundancy, and synergy), we generate synthetic data in a controlled environment based on the Trifeature dataset [22]. Furthermore, we assess the generalizability of InfMasking on several widely used multimodal benchmark datasets involving diverse modality combinations in real-world scenarios. These tasks span various domains (*e.g.*, healthcare, robotics, *etc.*) allowing for a thorough assessment of the model's representation capabilities across diverse modalities. Detailed experimental settings are provided in Appendix A. For evaluation, we use linear probing, *i.e.*, freezing the pre-trained feature extractor and training a linear classifier (or regressor, depending on the task) on top of the learned representations. The downstream task performance of the linear model serves as an indicator of the quality of the learned multimodal representations.

### 4.1 Synthetic Experiments on Trifeature Datasets

Following the experimental design of the Trifeature dataset in CoMM [12], we conduct controlled experiments on a synthetic dataset derived from Trifeature. We assess the model's capacity to learn uniqueness, redundancy and synergy through two separate experiments. In terms

of uniqueness and redundancy, we define shapes as redundant features and textures as uniqueness features. And the task involves two subtasks: (1) identifying the shared shape between two images (redundancy) and (2) predicting the texture of the first image (or second image) (uniqueness). The random-guessing baselines in both cases corresponds to 10%. As for synergy, we artificially introduce a strong correlation between textures and colors by defining a mapping $\mathcal{M}$ in the training set (*e.g.*, blue maps to triangles, stripes to red), resulting in a 50% baseline for random guessing. The model is trained on image pairs that follow this mapping. The task is to determine whether a given image pair satisfies the mapping $Y = \mathbb{1}(\text{texture}(X_1), \text{color}(X_2) \in \mathcal{M})$, thereby evaluating the model's ability to capture synergistic interactions across modalities.

Experimental results are illustrated in Tab. 1. Cross-modality constraints based on the InfoNCE loss [38] ("Cross" model) achieve the best performance at capturing redundant information but struggle with uniqueness and synergy. FactorCL [29], self-supervised constraints on each encoder ("Cross + Self" [51]) and MAE [19] (implementation details are provided in Appendix D.1) improve on uniqueness but remain limited in modeling synergy. CoMM [12] performs well across all three interactions. However, it still has considerable room for improvement, particularly in capturing synergistic information. In comparison, InfMasking achieves superior performance in capturing both redundancy and synergy, outperforming CoMM by 3.8% and 5.6%, respectively.

Table 1: Linear probing accuracy (in %) of redundancy (shape), uniqueness (texture) and synergy (color and texture) on Trifeature dataset. ♣ denotes results are from [12].

| Model | redundancy↑ | uniqueness↑ | synergy↑ |
|---|---|---|---|
| Cross♣ [38] | **100.0** | 11.6 | 50.0 |
| Cross+Self♣ [51] | 99.7 | 86.9 | 50.0 |
| FactorCL♣ [29] | 99.8 | 62.5 | 46.5 |
| MAE [19] | $99.8_{\pm 0.11}$ | $82.4_{\pm 3.09}$ | $50.1_{\pm 0.24}$ |
| CoMM [12] | $99.9_{\pm 0.06}$ | $86.8_{\pm 2.99}$ | $71.4_{\pm 3.47}$ |
| InfMasking (ours) | $99.9_{\pm 0.09}$ | $\mathbf{90.6}_{\pm 2.31}$ | $\mathbf{77.0}_{\pm 4.22}$ |

## 4.2 Experiments on Real-world Datasets

We further evaluate the performance of our model on several real-world multimodal datasets provided by Multibench [30]. These datasets span diverse modality combinations and task types, providing a comprehensive benchmark to assess the model's ability to learn effective multimodal representations. Further dataset details are provided in Appendix B.

### 4.2.1 Experiments with 2 Modalities on Multibench

Following the data preprocessing procedure of previous work [29, 12], we conduct our experiments using the same encoders, modality configurations and train models based on encoded inputs with diverse modalities. We consider "Cross", "Cross+Self", FactorCL and CoMM as baselines for comparison. As presented in Tab. 2, InfMasking consistently achieves the best performance across all

Table 2: Linear probing MSE($\times 10^{-4}$) for regression task and top-1 accuracy (in %) for classification tasks on Multibench. ♣ denotes results are from [29]. * denotes average is only selected from the results of classification tasks.

| Model | Regression | Classification | | | | |
|---|---|---|---|---|---|---|
| | V&T EE↓ | MIMIC↑ | MOSI↑ | UR-FUNNY↑ | MUSTARD↑ | **Average*** ↑ |
| Cross♣ [38] | $33.09_{\pm 3.67}$ | $66.7_{\pm 0.1}$ | $47.8_{\pm 1.8}$ | $50.1_{\pm 1.9}$ | $53.5_{\pm 2.9}$ | 54.52 |
| Cross+Self♣ [51] | $7.56_{\pm 0.31}$ | $65.49_{\pm 0.0}$ | $49.0_{\pm 1.1}$ | $59.9_{\pm 0.9}$ | $53.9_{\pm 4.0}$ | 57.07 |
| FactorCL♣ [29] | $10.82_{\pm 0.56}$ | $67.3_{\pm 0.0}$ | $51.2_{\pm 1.6}$ | $60.5_{\pm 0.8}$ | $55.80_{\pm 0.9}$ | 58.7 |
| CoMM [12] | $7.96_{\pm 2.13}$ | $66.4_{\pm 0.41}$ | $63.7_{\pm 2.5}$ | $63.3_{\pm 0.51}$ | $64.4_{\pm 1.1}$ | 64.45 |
| InfMasking (ours) | $\mathbf{4.23}_{\pm 0.37}$ | $\mathbf{68.1}_{\pm 0.42}$ | $\mathbf{69.0}_{\pm 1.2}$ | $\mathbf{64.3}_{\pm 0.9}$ | $\mathbf{66.8}_{\pm 2.5}$ | **67.05** |

benchmark datasets. In the binary classification tasks, InfMasking outperforms CoMM—the strongest baseline—by 1.7%, 5.3%, 1.0%, and 2.4% on the MIMIC, MOSI, UR-FUNNY, and MUSTARD datasets, respectively. For regression tasks, InfMasking also delivers superior results, achieving a

lead of $3 \times 10^{-4}$ in terms of MSE compared to the second-best model. These experimental results demonstrate the effectiveness of InfMasking in capturing bimodal interactions. Furthermore, its consistently strong performance across diverse datasets highlights the generalizability and robustness of InfMasking in real-world bimodal scenarios.

#### 4.2.2 Experiments with 3 Modalities on Multibench

We evaluate the generalizability of Inf-Masking in learning multimodal representations beyond two modalities. Specifically, we conduct experiments on two datasets: Vision&Touch (for the contact prediction task, with visual, force, and proprioceptive modalities) and UR-FUNNY (with visual, text, and audio modalities). CMC [41] and CoMM are selected as baselines for comparison in the three-modality setting.

The results are summarized in Tab. 3. To more intuitively assess the information gain introduced by incorporating a third modality, we additionally report results from bi-modal training scenarios using

Table 3: Linear probing top-1 accuracy (in %) for classification tasks on Vision&Touch and UR-FUNNY. ♣ denotes results are from [12].

| Model | #Mod. | V&T CP↑ | UR-FUNNY↑ |
|---|---|---|---|
| Cross | 2 | $86.3_{\pm 0.25}$ | $50.1^{\clubsuit}$ |
| Cross+Self | 2 | $87.6_{\pm 0.26}$ | $59.9^{\clubsuit}$ |
| CoMM | 2 | $87.0_{\pm 1.77}$ | $63.3_{\pm 0.51}$ |
| InfMasking (ours) | 2 | $88.5_{\pm 0.33}$ | $64.3_{\pm 0.9}$ |
| CMC$^{\clubsuit}$ [41] | 3 | 94.1 | 59.2 |
| CoMM | 3 | $\mathbf{94.1}_{\pm 0.17}$ | $64.8_{\pm 1.13}$ |
| InfMasking (ours) | 3 | $94.1_{\pm 0.09}$ | $\mathbf{65.6}_{\pm 1.15}$ |

CoMM, "Cross" and "Cross + Self". Specifically, we train these baselines on (1) the image and proprioceptive modalities of the Vision&Touch dataset, and (2) the image and text modalities of the UR-FUNNY dataset. Our experiments reveal that adding a third modality significantly enhances the performances of CoMM and InfMasking. CoMM as a strong baseline shows performance gains of 7.1% and 1.5% on Vision&Touch and UR-FUNNY, respectively. Although InfMasking's performance gain from adding the third modality is relatively modest compared to CoMM, it still matches CoMM's performance on the Vision&Touch dataset. On the UR-FUNNY dataset, InfMasking achieves the best result (+0.8%).

#### 4.2.3 Experiments with 2 Modalities on Multimodal IMDb

Multimodal IMDb(MM-IMDb) [2] is a real-world multimodal, multi-label dataset designed for movie genre classification. It poses two major challenges: significant class imbalance with genres such as comedy and drama dominating the label distribution, and substantial semantic discrepancy between visual (poster) and textual (plot's description) modalities. Since genre prediction based on a single modality

Table 4: Linear probing F1-score (weighted and macro) (in %) for MM-IMDB. △ indicates further training on unlabeded data. ♣ denotes results are from [12].

| Model | Modalities | weighted-f1↑ | macro-f1↑ |
|---|---|---|---|
| SimCLR$^{\clubsuit\triangle}$ [9] | V | $40.35_{\pm 0.23}$ | $27.99_{\pm 0.33}$ |
|  | V | 51.5 | 40.8 |
| CLIP$^{\clubsuit}$ [38] | L | 51.0 | 43.0 |
|  | V+L | 58.9 | 50.9 |
| SLIP$^{\clubsuit\triangle}$ [35] | V+L | $56.54_{\pm 0.19}$ | $47.35_{\pm 0.27}$ |
| CLIP$^{\clubsuit\triangle}$ [38] | V+L | $54.49_{\pm 0.19}$ | $44.94_{\pm 0.30}$ |
| CoMM$_{\text{(CLIP backbone)}}$ | V+L | $61.29_{\pm 0.73}$ | $53.79_{\pm 0.22}$ |
| InfMasking$_{\text{(ours, CLIP backbone)}}$ | V+L | $\mathbf{62.60}_{\pm 0.26}$ | $\mathbf{55.93}_{\pm 0.19}$ |

is often unreliable and the combination of both modalities can perform better [2], this underscores the need for effective modeling of multimodal interactions. We select both single-modal and multi-modal as baselines. For unimodal, we choose SimCLR (image-only) [9] and CLIP (pretrained on image-text pairs) [38]. For multimodal, we include CLIP, SLIP [35], and CoMM.

Tab. 4 summarizes the experimental results. Models trained on both modalities consistently outperformed their single-modality counterparts, further validating the importance of optimizing multimodal representation learning. InfMasking achieves the best overall performance, improving upon CoMM by 1.31% in weighted F1-score and 2.14% in macro F1-score. It is also worth noting that CLIP with

its original public weights achieves 58.9% on weighted F1-score, outperforming CLIP fine-tuned on MM-IMDb (54.59%). This suggests that redundant information learning may not always benefit complex tasks such as genre prediction, which require complementary modality alignment [12]. These results demonstrate the robustness and generalizability of InfMasking in handling imbalanced, semantically heterogeneous, and multi-label multimodal classification tasks.

## 5  Ablation Studies

To examine the effectiveness of the design of InfMasking, we conduct comprehensive ablation studies on the bimodal Trifeature dataset focusing on three critical components: the loss function formulation, the optimal number of masked views, and the masking ratio parameter.

Table 5: Linear probing accuracy (in %) of redundancy $R$, uniqueness $U$ and synergy $S$ on Trifeature Dataset under different combinations of loss functions. $\lambda_1$, $\lambda_2$, and $\lambda_3$ denote the weights for $\mathcal{L}_{mask}$, $\sum_i \mathcal{L}_i$, and $\mathcal{L}$, respectively, where $\mathcal{L}$ and $\sum_i \mathcal{L}_i$ correspond to the first and second terms in Eq. 1.

| loss weights | | | $R$ | $U_1$ | $U_2$ | $S$ | *Average* |
|---|---|---|---|---|---|---|---|
| $\lambda_1$ | $\lambda_2$ | $\lambda_3$ | | | | | |
| 0 | 0 | 1 | $95.8_{\pm1.91}$ | $85.9_{\pm2.11}$ | $83.8_{\pm2.97}$ | $58.7_{\pm7.11}$ | 80.1 |
| 0 | 1 | 1 | $99.9_{\pm0.06}$ | $87.1_{\pm3.31}$ | $86.5_{\pm2.60}$ | $71.4_{\pm3.47}$ | 86.0 |
| 1 | 1 | 0 | $\mathbf{99.9}_{\pm0.08}$ | $\mathbf{90.7}_{\pm2.10}$ | $\mathbf{91.4}_{\pm3.03}$ | $69.2_{\pm6.20}$ | 87.8 |
| 1 | 1 | 1 | $99.9_{\pm0.09}$ | $90.3_{\pm1.52}$ | $90.8_{\pm2.88}$ | $\mathbf{77.0}_{\pm4.22}$ | $\mathbf{89.5}$ |

**Loss function.** We conducted an ablation study on the Trifeature dataset to evaluate different loss combinations for capturing multimodal interactions. As shown in Tab. 5, the full objective ($\lambda_1 = \lambda_2 = \lambda_3 = 1$, InfMasking) achieves the highest synergy at 77.0% while maintaining balanced performance across other metrics. Using only CoMM loss ($\lambda_1 = 0$, $\lambda_2 = 1$, $\lambda_3 = 1$) yields 71.4% synergy, while using only $\mathcal{L}$ ($\lambda_1 = 0$, $\lambda_2 = 0$, $\lambda_3 = 1$) further decreases to 58.7%, indicating that CoMM loss alone is insufficient without the view diversity from masking. Excluding the $\mathcal{L}$(*i.e.*, $\hat{I}_{\mathrm{NCE}}(Z', Z'')$) drops synergy to 69.2%, despite marginal improvements in redundancy and uniqueness scores. As noted in CoMM [12], minimizing $\mathcal{L}$ enables the model to capture all information terms, albeit at a slower rate. When this loss is removed, the model learns redundancy and uniqueness more efficiently, achieving higher scores within the same epoch, but at the cost of diminished synergy performance.

**Number of masked views.** According to Section 3.2, increasing the number of masked views yields a closer approximation to $\mathbb{E}_{\mathrm{mask}}\left[\hat{I}_{\mathrm{NCE}}(Z'_{\mathrm{mask}}, Z')\right]$ and $\mathbb{E}_{\mathrm{mask}}\left[\hat{I}_{\mathrm{NCE}}(Z''_{\mathrm{mask}}, Z'')\right]$, albeit at a higher computational cost. As observed in Fig. 2(a), the synergy score improves progressively with an increasing number of views. Notably, performance is sufficiently robust when the number is in the range of $[6, 10]$, demonstrating practical feasibility for GPU implementation.

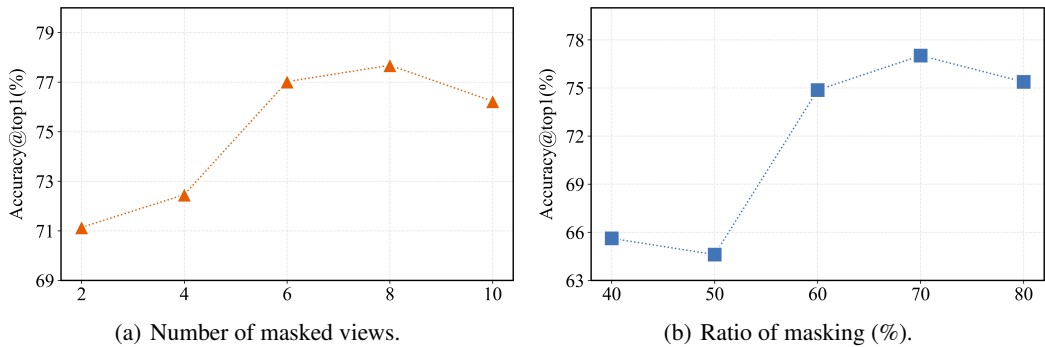

(a) Number of masked views.  (b) Ratio of masking (%).

Figure 2: Synergy accuracy changes with different masked setting on Trifeature datasets.

**Masking ratio.** Fig. 2(b) illustrates the impact of varying the masking ratio. At lower ratios ($\leq 50\%$), although the model can capture synergistic information, the overall performance remains

unsatisfactory. In contrast, maintaining a higher masking ratio enables the model to generate superior multimodal representations that effectively leverage complementary information across modalities. Furthermore, a higher masking ratio can also provide a greater speedup benefit [19].

## 6 Related Work

**Multimodal learning.** Multimodal learning integrates diverse data sources—such as text, image, audio, and tactile inputs—to enhance information understanding across modalities [31, 36, 17, 32, 10]. Traditional approaches rely on simple fusion techniques like feature concatenation [11] or modality-specific prediction averaging [14]. Transformer architectures revolutionized this field through dynamic cross-modal attention mechanisms [42, 50]. Contemporary approaches typically follow a two-stage framework: training specialized encoders for each modality, then projecting these representations into a unified embedding space [4, 12]. This paradigm has been applied across representation learning [5, 38], cross-modal alignment [28, 21], and generative modeling [1, 39].

**Self-Supervised multimodal representation learning.** Self-supervised learning generates supervision signals from data's inherent structure [13, 24, 48, 47]. In multimodal contexts, these approaches leverage cross-modal correspondences [53, 45]. Prior methods have explored generative approaches, such as reconstructing one modality from another [39], and masked prediction strategies for joint modality modeling [3]. Contrastive learning has emerged as particularly effective for multimodal representation learning [15, 38], constructing positive pairs through data augmentation [9] and introducing both intra-modal and cross-modal objectives [34]. Some approaches incorporate regularization terms to align representations across modalities [44].

**Contrastive Multimodal Interactions.** Contrastive multimodal approaches [38, 23] optimize cross-modal contrastive loss but emphasize redundant information while neglecting unique and synergistic information requiring joint consideration, with FactorCL [29] addressing this through explicit modeling despite practical implementation challenges. CoMM [12] advances the field using multimodal augmentations and information theory-grounded losses to capture various interaction patterns, though enhancing synergistic interactions remains challenging.

## 7 Conclusion

This paper introduces InfMasking, a contrastive method that effectively captures synergistic information in multimodal representation learning by stochastically occluding features during fusion and aligning representations through mutual information maximization. We derive a computationally efficient approximation for infinite masking patterns and demonstrate that our approach not only enhances synergistic information extraction in controlled settings but also achieves state-of-the-art performance across seven diverse multimodal benchmarks. InfMasking has some limitations. It lacks a rigorous theoretical framework to systematically analyze the mechanisms of synergistic interactions. Future research will prioritize developing comprehensive theoretical foundations to formally characterize and measure the synergistic information.

### Acknowledgments

This work was supported by the National Natural Science Foundation of China (No. U24A20323), the Major Science and Technology Special Project of the Sichuan Provincial Department of Science and Technology (Grant No. 2024ZDZX0002), the Sichuan Provincial Innovation Group Project (Grant No. 2024NSFTD0054), Fundamental Research Funds for the Central Universities (JBK202511081), Blockchain Research Center of China, Natural Science Foundation of China under Grant No. 62502397, National Natural Science Foundation of China (NSFC) (72471197), and the Sichuan Provincial Philosophy and Social Science Fund (Grant No. SCJJ25ND091) . We thank anonymous reviewers for their insightful feedback and Benoit Dufumier, the first author of CoMM [12], for his contributions to open sourcing the code.

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

# Appendix

## A  Experimental Details

**Training protocol.**  All experiments are conducted using five independent runs with random seeds in the range $[42, 46]$. We report the mean and standard deviation of performance metrics (*i.e.*, accuracy, mean squared error) to account for variability across runs. Early stopping based on validation accuracy is systematically applied to prevent overfitting. The best-performing checkpoint on the validation set is selected for final evaluation on the test set.

For dataset-specific encoder architectures, modality-specific data augmentation and latent converters, we follow the same configurations as CoMM [12].

**Training details.**  We use AdamW [33] as the optimizer in all experiments. Detailed hyperparameters are listed in Tab. 6. Following [12] on MM-IMDb, we also use a cosine scheduler with final value $10^{-6}$ and a warmup over 10 epochs. And all models are trained for 100 epochs except for MM-IMDb which is trained for 70 epochs. All experiments are conducted on a single NVIDIA 4090 GPU with 24GB memory.

Table 6: Hyperparameters for InfMasking. *Masking ratio* is the ratio of masking for each masked view. The V&T CP and V&T EE are the contact prediction and end-effector position prediction tasks on Vision&Touch dataset respectively.

| dataset | learning rate | masking ratio | number of masked views |
|---|---|---|---|
| *Trifeature* | $3 \times 10^{-4}$ | 0.7 | 6 |
| *MIMIC* | $3 \times 10^{-4}$ | 0.8 | 6 |
| *MOSI* | $1 \times 10^{-3}$ | 0.8 | 5 |
| *UR-FUNNY*(2 modalities) | $1 \times 10^{-3}$ | 0.5 | 4 |
| *MUSTARD* | $1 \times 10^{-3}$ | 0.5 | 5 |
| *V&T CP*(2 modalities) | $1 \times 10^{-4}$ | 0.7 | 6 |
| *V&T EE* | $1 \times 10^{-4}$ | 0.5 | 4 |
| *MM-IMDb* | $1 \times 10^{-3}$ | 0.8 | 4 |
| *UR-FUNNY*(3 modalities) | $1 \times 10^{-3}$ | 0.5 | 4 |
| *V&T CP*(3 modalities) | $1 \times 10^{-4}$ | 0.8 | 5 |

**Fusion module configuration.**  For all experiments involving InfMasking, we employ the fusion module similar to that used in CoMM [12], which operates on a sequence of modality-specific embeddings and is implemented as a Transformer-based encoder layer. Specifically, the architecture consists of multi-head self-attention followed by a feed-forward network, with residual connections and layer normalization. In the bimodal setting, we use a 1-layer Transformer with 8 attention heads, while in the trimodal setting, a 2-layer Transformer with the same number of heads is adopted. In addition, a learnable `[CLS]` token is appended to the input sequence, which serves as a global representation aggregating information across modalities.

## B  Dataset Details

### B.1  Trifeature

The Trifeature dataset [22] is designed to investigate the properties of visual neural networks and comprises three distinct features: shape, color, and texture. Each feature consists of 10 categories, resulting in 1,000 unique combinations. Of these, 800 are used for training and 200 for testing. Each training combination is instantiated three times with random rotations applied to both shape and texture components. Shapes are rendered within a $128 \times 128$ bounding box, with rotation angles uniformly sampled from $[-45°, 45°]$, and then randomly placed within a $224 \times 224$ image canvas while ensuring full visibility. Texture and color are independently applied in the same manner. Image pairs are constructed from these instances, resulting in 10,000 training pairs and 4,096 test pairs, both sampled from the same underlying distribution.

### B.2 Multibench

According to [30], all datasets below have been pre-processed to ensure the removal of any personally identifiable information and to safeguard user privacy (some datasets don't include any personal information, *e.g.* Vision&Touch and MM-IMDb).

- **MIMIC** [25] comprises 53,423 hospital admissions from 38,597 distinct patients, spanning the years 2001 to 2012. It includes two modalities: hourly clinical measurements over a 24-hour period (represented as 12-dimensional vectors, times series modality) and static patient information such as age and gender (represented as 5-dimensional vectors, tabular modalities). The task is a binary classification problem aiming to predict whether a patient belongs to ICD-9(*International Statistical Classification of Diseases and Related Health Problems*) code group 7 (460–519), which is commonly used in studies on disease classification [29].

- **MOSI** [52] consists of 2,199 video clips collected from YouTube, designed for sentiment analysis tasks. Each sample includes video, audio signals, and corresponding text transcriptions. The original annotations range continuously from -3 to 3; following the approach in [29], these labels are binarized into positive and negative classes. The model is trained based on textual and visual modalities.

- **UR-FUNNY** [18] is constructed from 1,866 TED talk videos and comprises 16,514 samples for the task of humor detection. Each sample contains video, audio, and corresponding text transcriptions. The objective is to determine whether a given sequence is humorous, formulated as a binary classification problem. For the bi-modal setting, we use the textual and visual modalities.

- **MUSTARD** [8] is designed for sarcasm detection and is sourced from television shows such as Friends. It contains 690 balanced utterances, each comprising video, audio, and text transcriptions, annotated as either sarcastic or non-sarcastic. In our experiments, we utilize the textual and visual modalities.

- **Vision&Touch** [27] comprises data from robotic manipulation tasks, consisting of 150 trajectories, each with 1,000 time steps. The dataset includes RGB images, depth maps, force measurements, and end-effector positions and velocities. The benchmark tasks are (1) binary classification to predict whether contact will occur in the next step and (2) regression to predict the end-effector position, evaluated using mean squared error (MSE). For the bi-modal setting, we use the visual and proprioceptive modalities.

### B.3 MM-IMDb

Multimodal IMDb (MM-IMDb) [2] is designed for movie genre prediction and comprises 25,959 films, each annotated with posters, plot summaries, genre labels, and metadata. Derived from the Movielens 20M dataset [16], this benchmark focuses on 23-way multi-label classification. In our experiments, we utilize the image modality (movie posters) and the text modality (plot summaries). While MM-IMDb is part of the Multibench benchmark [30], we present it separately in our experiments, as our model is trained directly on the raw data instead of relying on the pre-processed features offered by Multibench.

## C  Broader Impact

This study aims to enhance the modelling of cross-modal synergy to generate more informative multimodal representations. To ensure that InfMasking can be deployed responsibly in real-world scenarios, we highlight several key considerations.

**Computational complexity.** InfMasking introduces an infinite masking strategy that maximizes the mutual information between masked fused views and their unmasked counterparts, strengthening the complementarity of different modalities. However, this method inevitably increases GPU memory usage, as each additional masked view amplifies the memory footprint. We encourage future work to explore lightweight variants that can alleviate the associated computational demands.

**Privacy and security.** As discussed in Section 4.2.1, the datasets used in this study span multiple domains, including healthcare, sentiment analysis and multimedia. According to [30], all instances

containing personal information have been rigorously anonymized and de-identified. And the Vision&Touch and MM-IMDb datasets do not contain any personally identifiable information. All experiments are conducted using irreversible, pre-extracted features, except for MM-IMDb, which is processed directly from raw data; no raw or reconstructable user data is accessed, thereby minimizing privacy risks.

**Future work.** Future research will focus on establishing rigorous theoretical frameworks to quantify and formally characterize the synergistic information extracted by InfMasking. Such frameworks would provide mathematical guarantees on information preservation while elucidating the fundamental limits of multimodal representation learning. Additionally, we aim to develop adaptive masking strategies that dynamically optimize masking patterns based on task requirements and modality-specific characteristics, potentially employing reinforcement learning to fine-tune these configurations. These advancements would significantly enhance our capacity to model complex synergistic relationships in multimodal data, advancing the field toward more generalizable multimodal intelligence.

# D  Additional Experiments

## D.1  Difference with MAE

Masked Autoencoders(MAE) [19] achieve self-supervised learning through reconstruction of masked image patches, which consist of two parts. For the encoder, it encodes randomly masked image patches into latent features. The decoder is trained to predict the masked patches using reconstruction loss, thereby enhancing semantic relationships between them in a single-modal (vision) setting. InfMasking adapts and extends this masking paradigm to a contrastive multimodal context. It focuses on aligning and extracting synergistic information from multimodal tokens (*e.g.*, features from text, images, audio, or tabular data) through infinite masking, emphasizing cross-modal interactions like redundancy, uniqueness, and synergy.

Table 7: Linear probing accuracy (%) on three datasets from MultiBench [30] for MAE, CoMM, and InfMasking models.

| Dataset | MIMIC | UR-FUNNY | MOSI | average |
|---------|-------|----------|------|---------|
| CoMM | $66.4_{\pm 0.41}$ | $63.3_{\pm 0.51}$ | $63.7_{\pm 2.5}$ | 64.47 |
| MAE | $67.4_{\pm 0.3}$ | $62.5_{\pm 1.43}$ | $65.4_{\pm 1.6}$ | 65.1 |
| InfMasking | $68.1_{\pm 0.42}$ | $64.3_{\pm 0.9}$ | $69.0_{\pm 1.2}$ | 67.12 |

Unlike MAE, our masking approach does not mask the raw input of each modality but rather masks the features of each modality before fusion. Furthermore, we aim to maximize mutual information between masked and unmasked multimodal representations without reconstruction. It derives a lower bound approximation for the InfMasking loss assuming Gaussian distributions for masked features, making it computationally feasible for infinite views. This makes InfMasking a natural evolution for handling diverse modalities, addressing limitations in traditional contrastive learning (*e.g.*, over-reliance on multiview redundancy) while preserving MAE's core idea of using masking to create challenging, informative views.

We further compare our InfMasking loss with MAE reconstruction loss on multiple datasets from MultiBench [30]. The results are illustrated in Tab. 7. Under the same experimental conditions, we randomly mask tokens across modalities, encoder forward pass with masked iuput, and decoder-based reconstruction focused solely on masked tokens (using MSE loss averaged over masked positions). This creates a generative baseline analogous to MAE but extended to multimodal tokens. The MAE variant replaces the InfMasking loss component terms in Eq. (2) with reconstruction loss.

## D.2  Ablation Studies of Data Augmentation on Trifeature Datasets

The InfoMin Principle [41] plays a pivotal role in self-supervised learning. It demonstrates that data augmentation is an effective strategy for adhering to this principle, as stronger data augmentations reduce mutual information to an optimal level. In our work, we adopt the same settings for modality-specific data augmentation as outlined in CoMM [12]. To extend this investigation, we further explore the influence of data augmentation strategies on the bimodal Trifeature dataset [22].

Table 8: Impact of data augmentation on linear probing accuracy (%) for multimodal interactions. The term "All" refers to SimCLR [9] augmentations. InfMasking applies "All" augmentations to both modalities, consistent with CoMM.

| Augmentations | | $R$ | $U_1$ | $U_2$ | $S$ | Average |
|---|---|---|---|---|---|---|
| Modality 1 | Modality 2 | | | | | |
| {All} | $\varnothing$ | $99.78_{\pm0.08}$ | $85.28_{\pm2.88}$ | $49.89_{\pm8.73}$ | $50.0_{\pm0.0}$ | 71.24 |
| $\varnothing$ | {All} | $99.85_{\pm0.06}$ | $49.08_{\pm2.65}$ | $87.44_{\pm3.59}$ | $50.0_{\pm0.0}$ | 71.59 |
| {All} \{crop} | {All} | $97.70_{\pm0.84}$ | $58.07_{\pm2.68}$ | $87.15_{\pm3.80}$ | $50.0_{\pm0.0}$ | 73.23 |
| {All} | {All} \{crop} | $96.91_{\pm1.99}$ | $85.42_{\pm4.01}$ | $57.85_{\pm6.63}$ | $50.0_{\pm0.0}$ | 72.54 |
| InfMasking | | $99.86_{\pm0.10}$ | $90.30_{\pm1.52}$ | $90.80_{\pm2.88}$ | $77.02_{\pm4.22}$ | 89.5 |

As shown in Tab. 8, omitting data augmentation leads to a significant degradation in model performance, particularly in uniqueness. Notably, cropping as a critical transformation in self-supervised learning for vision tasks [9, 20], is vital for learning synergistic representations in the Trifeature dataset. When modality-specific cropping augmentation is omitted, the model struggles to capture the uniqueness of the corresponding modality, resulting in an inability to effectively learn synergy.

# E    Analysis of Gaussian Approximation Assumption via Visualization

Based on the theoretical framework of InfMasking discussed in Sec. 3.2, we further analyze the robustness of Gaussian approximation assumption through visualization.

We employ dimensionality reduction to project the high-dimensional embeddings of multimodal features and their masked view features from the Trifeature dataset into a two-dimensional space, which are visualized using t-SNE. As shown in Fig. 3, the masked embeddings cluster around a central point in the projected space. Notably, this central point aligns closely with the multimodal features, indicating that despite the perturbations introduced by masking, the masked embeddings inherently preserve core aspects of the synergistic semantic nature. It uncovers the stability of synergistic integration: even when parts of modalities are obscured, the fused representations converge toward a shared semantic manifold, reflecting the emergent properties that arise from modal complementarity rather than redundancy alone. Furthermore, the dispersion of masked embeddings similarly indicates semantic differentiation within the ambient embedding space. The observed variance highlights subtle nuances in how masking affects the differences of representations combined from masked view features, potentially corresponding to variations in semantic granularity—such as implicit biases or contextual implications that become apparent only through joint modal analysis.

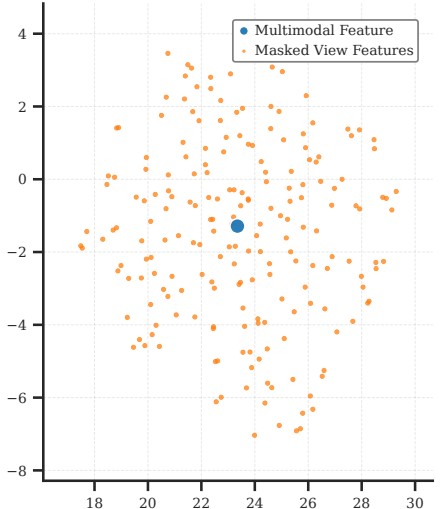

Figure 3: Visualization of the distribution of multimodal fusion embeddings and its masked counterpars.

# F    Pseudo-Code

Algorithm 1 outlines the training procedure of InfMasking, formulated in the general case with $n$ modalities (*e.g.*, image, text, audio, *etc*).

The key input components are as follows: $\mathcal{T}^\star$ denotes a set of label-preserving transformations used for data augmentation. The fusion transformer $g$ integrates latent features from diverse modalities. The masked view number $M'$ indicates how many masked instances are generated per modality. The random masking operator $\mathcal{M}$ stochastically obscures portions of the embedding features. And the temperature parameter $\tau$ controls the sharpness of the total loss.

**Algorithm 1** Multimodal contrastive interaction learning with InfMasking

---

1: **Input:** Multi-modal dataset $\{X_1, X_2, ..., X_n\}$, label-preserving transformations $\mathcal{T}^\star$, set of projection transformations $\mathcal{P} = \{p_1, \ldots, p_n\}$, batch size $N$, masked view number $M'$, uni-modal encoders $(f_i)_{i \in [1..n]}$, fusion transformer $g$, random mask operator $\mathcal{M}$, temperature parameter $\tau$.

2: **for** sampled mini-batch $\{\mathbf{x}_k\}_{k \in [1..N]} = (\mathbf{x}_k^1, ..., \mathbf{x}_k^n)_{k \in [1..N]}$ **do**

3:      **for** $k \in [1..N]$ **do**

4:          draw $t', t'' \sim \mathcal{T}^\star$

5:          $\mathbf{x'}_i, \mathbf{x''}_i \leftarrow t'(\mathbf{x}_i), t''(\mathbf{x}_i)$

6:          **for** $j \in [1..M']$ **do**

7:              $\mathbf{z'}_{mask}^j \leftarrow g\left(\mathcal{M}(f_1(\mathbf{x'}_k^1)), ..., \mathcal{M}(f_n(\mathbf{x'}_k^n))\right)$

8:              $\mathbf{z''}_{mask}^j \leftarrow g\left(\mathcal{M}(f_1(\mathbf{x''}_k^1)), ..., \mathcal{M}(f_n(\mathbf{x''}_k^n))\right)$

9:          **end for**

10:          $\mathbf{z'}_k \leftarrow g(f_1(\mathbf{x'}_k^1), ..., f_n(\mathbf{x'}_k^n))$

11:          $\mathbf{z''}_k \leftarrow g(f_1(\mathbf{x''}_k^1), ..., f_n(\mathbf{x''}_k^n))$

12:          **for** $i \in [1..n]$ **do**

13:              $\mathbf{x}_k^i \leftarrow p_i(\mathbf{x}_k)$

14:              $\mathbf{z}_k^i \leftarrow g(f_i(\mathbf{x}_k^i))$

15:          **end for**

16:      **end for**

17:      $\mathcal{L}_{\text{InfMasking}} \leftarrow -\frac{1}{M'} \sum_{k=1}^{M'} \left[ \mathbb{E}_{z'_{mask}^k, z'_{\text{pos}} \sim p(Z'_{mask}^k, Z')} \left[ \log \frac{\exp(z'_{mask}^{k\,T} z'_{\text{pos}}/\tau)}{\exp(z'_{mask}^{k\,T} z'_{\text{pos}}/\tau) + \sum_{z'_{\text{neg}}} \exp(z'^T z'_{neg}/\tau)} \right] \right.$

18:          $\left. + \mathbb{E}_{z''_{mask}^k, z''_{\text{pos}} \sim p(Z''_{mask}^k, Z'')} \left[ \log \frac{\exp(z''_{mask}^{k\,T} z''_{\text{pos}}/\tau)}{\exp(z''_{mask}^{k\,T} z''_{\text{pos}}/\tau) + \sum_{z''_{\text{neg}}} \exp(z''^T z''_{neg}/\tau)} \right] \right]$

19:      **for** $i \in [1..n]$ **do**

20:          $\mathcal{L}_i \leftarrow -\left[ \mathbb{E}_{z_i, z'_{\text{pos}} \sim p(Z_i, Z')} \left[ \log \frac{\exp(z_i^T z'_{\text{pos}}/\tau)}{\exp(z_i^T z'_{\text{pos}}/\tau) + \sum_{z'_{\text{neg}}} \exp(z_i^T z'_{\text{neg}}/\tau)} \right] \right.$

21:          $\left. + \mathbb{E}_{z_i, z''_{\text{pos}} \sim p(Z_i, Z'')} \left[ \log \frac{\exp(z_i^T z''_{\text{pos}}/\tau)}{\exp(z_i^T z''_{\text{pos}}/\tau) + \sum_{z''_{\text{neg}}} \exp(z_i^T z''_{\text{neg}}/\tau)} \right] \right]$

22:      **end for**

23:      $\mathcal{L} \leftarrow -\mathbb{E}_{z', z''_{\text{pos}} \sim p(Z', Z'')} \left[ \log \frac{\exp(z'^T z''_{\text{pos}}/\tau)}{\exp(z'^T z''_{\text{pos}}/\tau) + \sum_{z''_{\text{neg}}} \exp(z'^T, z''_{\text{neg}}/\tau)} \right]$

24:      $\mathcal{L}_{\text{Total loss}} \leftarrow \mathcal{L} + \sum_{i=1}^n \mathcal{L}_i + \mathcal{L}_{\text{InfMasking}}$

25:      update $(f_i)_{i \in [1..n]}, \mathcal{M}, g$ to minimize $\mathcal{L}_{\text{Total loss}}$

26: **end for**

27: **return** $(f_i)_{i \in [1..n]}, g$

---

# G Proof

**Proof 1 (lemma 3)**

$$\mathbb{E}_{mask}[\hat{I}_{NCE}(Z'_{mask}, Z')] \tag{8}$$

$$= \mathbb{E}_{mask}[\mathbb{E}_{z'\sim p(Z')} \left[ \log \frac{\exp(z'^T z'_{mask}/\tau)}{\exp(z'^T z'_{mask}/\tau) + \sum_{z'_{neg}} \exp(z'^T z'_{neg}/\tau)} \right]] \tag{9}$$

$$= \mathbb{E}_{z'\sim p(Z')}[\mathbb{E}_{mask} \left[ \log \frac{\exp(z'^T z'_{mask}/\tau)}{\exp(z'^T z'_{mask}/\tau) + \sum_{z'_{neg}} \exp(z'^T z'_{neg}/\tau)} \right]] \tag{10}$$

$$= \mathbb{E}_{z'\sim p(Z')}[\mathbb{E}_{mask} \left[ (z'^T z'_{mask}/\tau) - \log[\exp(z'^T z'_{mask}/\tau) + \sum_{z'_{neg}} \exp(z'^T z'_{neg}/\tau)] \right]] \tag{11}$$

$$\geq \mathbb{E}_{z'\sim p(Z')} \left[ z'^T \mathbb{E}_{mask}[z'_{mask}]/\tau - \log \mathbb{E}_{mask}[\exp(z'^T z'_{mask}/\tau) + \sum_{z'_{neg}} \exp(z'^T z'_{neg}/\tau)] \right] \tag{12}$$

*The inequality Eq.(12) merges from the application of Jensen inequality on concave functions i.e.,*
*$\mathbb{E}_x \log(X) \leq \log \mathbb{E}_x[X]$. $z'_{mask}$ denotes the integrated representation derived from the fusion of all*
*masked features across diverse modalities via the Transformer architecture.*

**Lemma 4** *Consider a random variable $x$ that follows a multivariate Gaussian distribution, denoted*
*as $x \sim \mathcal{N}(\mu, \Sigma)$, where $\mu \in \mathbb{R}^n$ represents the mean vector and $\Sigma \in \mathbb{R}^{n \times n}$ is the covariance matrix*
*The moment generating function (MGF) of this random variable is given by the following expression:*

$$\mathbb{E}_x \left[ e^{a^T x} \right] = e^{a^T \mu + \frac{1}{2} a^T \Sigma a}, \tag{13}$$

*where $a \in \mathbb{R}^n$ is an arbitrary constant vector.*

*$z'^T z'_{mask}/\tau$ According to Lemma 4, we can derive the MGF of the inequality Eq.(7) as follows:*

$$\mathbb{E}_{mask}[\hat{I}_{NCE}(Z'_{mask}, Z')] \tag{14}$$

$$\geq \mathbb{E}_{z'\sim p(Z')} \left[ z'^T \mu_{z'_{mask}}/\tau - \log[\exp(z'^T \mu_{z'_{mask}}/\tau + \frac{1}{2\tau^2} z'^T \Sigma_{z'_{mask}} z) + \sum_{z'_{neg}} \exp(z'^T z'_{neg}/\tau)] \right]$$
$$\tag{15}$$

