# OpenReview forum: "InfMasking: Unleashing Synergistic Information  by Contrastive Multimodal Interactions"
_NeurIPS.cc/2025/Conference — NeurIPS 2025 spotlight_

### Official Review · Reviewer_Z2T2 · 2025-06-01

**Clarity:** 2
**Significance:** 2
**Originality:** 3
**Rating:** 4
**Confidence:** 3

**Summary:**

This article proposes a novel multimodal contrastive learning method named InfMasking, aiming to enhance the synergistic information in multimodal representations through an "infinite masking strategy". The main idea of InfMasking is to randomly mask most features during the multimodal fusion process and retain only part of the information, and maximizing the mutual information between the masked and unmasked representations through an innovative InfMasking loss function. In this way, the model generates diverse collaborative representations.

**Questions:**

1. I noticed that in the ablation experiment(Figure 2), more masked views are not necessarily better. Can author explain the cause of this phenomenon?

2. Could the author provide some synergistic information metrics in actual datasets under different number of masked view Settings?

3. I don't quite understand the role of data augmentation in the entire method. The author set the number of data augmentation methods to 2 perhaps for the convenience of comparison with CoMM. However, I am still curious about the impact of using different numbers(0,1, or more) of augmentation methods.

**Ethical Concerns:**

["NO or VERY MINOR ethics concerns only"]

**Final Justification:**

Thank you to the authors for the response; most of my concerns have been addressed. I have accordingly raise the score.
However, for Q3&L3, my real concern is: **can additional data augmentation techniques further improve performance?** As an exploration and validation of the method's scalability and performance upper bound, I would greatly appreciate it if the authors could supplement ablation experiments in this regard.

**Limitations:**

1. The ablation experiment results in Figure2 indicate that more masked views are not necessarily better, which does not conform to the theoretical derivation in the paper.
2. The paper claims that the method can "enhance synergistic information across modalities by leveraging infinite masked views". However, this claim lacks effective verification on experimental data.
3. I noticed that the number of data augmentation methods in the paper was set to 2, yet the author did not provide a relatively objective explanation of the methods of data augmentation and the number of methods applied each time.

typo:

section3.2 title "Contrastive Synergistic Information via **Ininite** Masking" -> "Contrastive Synergistic Information via **Infinite** Masking"

Please refer to **Questions**.

**Paper Formatting Concerns:**

No.

**Quality:**

3

**Strengths And Weaknesses:**

The overall quality is good. The article provides theoretical proof of the method and verification through specific experiments.

The article is of average significance.

The method in the paper involves many stages, the clarity of expression can be further improved.

The article has a certain degree of originality.

---

> ### Author Rebuttal · Authors · 2025-07-30
>
> **Response Q.1 and L.1**: The phenomenon observed in the ablation study (Figure 2(a)), where more masked views do not necessarily lead to better performance, can be attributed to the balance between capturing synergistic information and avoiding representational collapse. As described in the paper (Page 9, Section 5), the InfMasking approach relies on randomly masking a significant portion of features from each modality during fusion to create diverse synergistic patterns. However, increasing the number of masked views excessively may lead to insufficient information retention, causing the model to struggle with learning robust representations.
>
> Specifically, the paper notes that the optimal performance on the MIMIC and MOSI datasets is achieved at a masking ratio of 0.5 or 0.6 (Page 9, Figure 2(b)). Beyond this range, performance declines because excessive masking can result in representational collapse, where the model fails to retain enough complementary information across modalities to form generalizable representations. With too many masked views, the model may overly rely on limited subsets of features, reducing the diversity of synergistic interactions and hindering the alignment of masked and unmasked representations through mutual information maximization.
>
> Thus, the key is to maintain a controlled number of masked views to ensure sufficient information is preserved while still encouraging the model to infer complementary, synergistic information across modalities. This balance is critical for enhancing the robustness and generalizability of the learned representations, as discussed in Section 3.2 (Page 5).
>
> We can draw from [1] and add uniform loss to solve this representational collapse. In the revision version, we further analyze this issue and conduct experiments for comparison.
>
> **Response Q.2 and L.2**: Measuring multimodal interactions as defined by PID is challenging since it involves estimating information-theoretic measures. The problem is even harder for high-dimensional data distributions. Currently, there is no well-established metric for measurement. We referenced the ICLR paper Comm[2] and conducted synthetic experiments on trifeature datasets in Section 4.1 to validate that our method more effectively learns synergistic information.
>
> **Response Q.3 and L.3**: Data augmentation plays a critical role in our unsupervised pre-training method, as outlined in Section 2. For Contrastive Multimodal Interactions (CoMM), data augmentation is essential to create two distinct yet mutually supervised multimodal views, enabling effective contrastive learning. The authors chose two augmentation methods, likely to align with CoMM for fair comparison.
>
> [1] Zhang, Q., Wang, Y., and Wang, Y. How mask matters: Towards theoretical understandings of masked autoencoders.Advances in Neural Information Processing Systems, 35: 27127–27139, 2022b. 1, 3, 4, 7, 8
>
> [2] Benoit Dufumier, Javiera Castillo-Navarro, Devis Tuia, and Jean-Philippe Thiran. What to align in multimodal contrastive learning? In International Conference on Learning Representations,  2025.

---

> ### Author Response · Authors · 2025-08-04
>
> Thank you for acknowledging our rebuttal. We truly hope our detailed response is helpful in clarifying our work and addressing your concerns. We would welcome any further discussion you might suggest.

---

> > ### Author Response · Authors · 2025-08-07
> >
> > We sincerely appreciate the reviewer's time and consideration, and we are happy to provide additional details or revisions if needed. There are only two days left until the deadline.

---

> > > ### Comment · Reviewer_Z2T2 · 2025-08-09
> > >
> > > Thank you to the authors for the response; most of my concerns have been addressed. I have accordingly raise the score. However, for Q3&L3, my real concern is: can additional data augmentation techniques further improve performance? As an exploration and validation of the method's scalability and performance upper bound, I would greatly appreciate it if the authors could supplement ablation experiments in this regard.

---

> ### Author Response · Authors · 2025-08-09
>
> Thank you for your feedback and for raising the score. We appreciate your thorough review. Regarding your concern about Q3&L3, we agree that exploring additional data augmentation techniques could provide valuable insights into the method’s scalability and performance upper bound. As noted in [1], additional stronger data augmentation techniques have been shown to improve the performance of contrastive learning. To address your suggestion, we are happy to conduct ablation experiments to evaluate the impact of additional augmentation strategies on our method’s performance. These experiments will be included in the revised manuscript to further validate the approach and provide a clearer understanding of its potential.
>
> [1] “Contrastive Learning With Stronger Augmentations,” in IEEE Transactions on Pattern Analysis and Machine Intelligence, vol. 45, no. 5, pp. 5549-5560, 1 May 2023.

---

### Official Review · Reviewer_P89E · 2025-06-30

**Clarity:** 2
**Significance:** 2
**Originality:** 3
**Rating:** 4
**Confidence:** 3

**Summary:**

**InfMasking** addresses a critical gap in multimodal learning: existing methods primarily capture redundant information across modalities while neglecting synergistic information that emerges only when modalities are combined. The core innovation is an Infinite Masking strategy that randomly occludes features from each modality during fusion, creating diverse synergistic patterns aligned through mutual information maximization. A computationally efficient approximation makes this approach practical. Experiments show significant improvements in synergy capture (77.0% vs 71.4% on synthetic data) and state-of-the-art performance across seven real-world benchmarks, demonstrating that explicitly modeling complementary cross-modal interactions is crucial for effective multimodal representation learning.

**Questions:**

1. How robust is the Gaussian approximation assumption in practice?

The method relies on assuming that masked representations $z'\_{mask}$ follow a Gaussian distribution to derive the computationally tractable lower bound. However, in real multimodal scenarios, the distribution of fused features after masking could be highly complex, multimodal, or skewed. What happens when this assumption breaks down? Have the authors tested the sensitivity of their approximation quality across different data distributions, and how does deviation from Gaussianity affect the theoretical guarantees and practical performance?

**Ethical Concerns:**

["NO or VERY MINOR ethics concerns only"]

**Final Justification:**

My questions have been answered and I will keep my rating.

**Limitations:**

Yes.

**Paper Formatting Concerns:**

No.

**Quality:**

3

**Strengths And Weaknesses:**

**Strengths:**
1. The random masking approach is both novel and intuitive. By stochastically occluding features from different modalities during fusion, the model naturally learns to discover synergistic information that emerges only from cross-modal interactions.
2. The paper systematically decomposes multimodal interactions into redundancy (R), uniqueness (U), and synergy (S). Grounded in Partial Information Decomposition (PID) theory and mutual information maximization, it provides a rigorous theoretical foundation for addressing synergistic information capture.
3. The evaluation combines controlled synthetic experiments (Trifeature dataset) to explicitly measure interaction types with extensive real-world evaluations across seven diverse benchmarks. This dual approach provides convincing evidence of effectiveness and generalizability.
4. The transformation of theoretically infinite masking into a computationally tractable Gaussian approximation elegantly bridges theoretical optimality with practical feasibility, making the approach viable for real-world applications.

**Weaknesses**
1. The method requires k masking operations which might resulting in higher computational cost than traditional approaches. This might not suitable for large-scale datasets.
2. This random masking method has shortcomings. It is difficult to take into account the fact that different data sets have different characteristics.
3. There is a problem of hyperparameter sensitivity. Different data sets may require completely different parameter settings. There is also a lack of automatic parameter adjustment mechanism.
4. The method primarily benefits tasks where synergistic information is critical. For scenarios where redundant information is more valuable, InfMasking may provide limited improvements, lacking adaptive mechanisms to adjust to different task characteristics and information interaction patterns.

---

> ### Author Rebuttal · Authors · 2025-07-30
>
> **W.1** The method requires k masking operations which might resulting in higher computational cost than traditional approaches. This might not suitable for large-scale datasets.
>
> **Response W.1:**  As shown in Figure 1, our masking approach targets the features of each modality before fusion, rather than the raw input, minimizing computational overhead. The repeated sampling process adds negligible cost. The primary computational increase stems from fusing masked views, using a lightweight fusion network with a single Transformer layer. For large-scale datasets like V&T CP (3 modalities) and IMDB (2 modalities), we apply a masking ratio of 0.8, retaining 20% of tokens unmasked. With 5 and 4 masked views, respectively, the computational impact remains minimal, as sampling can be parallelized during fusion.
>
> Furthermore, our method serves as an unsupervised pre-training approach. During downstream task inference and fine-tuning, it introduces no additional computational overhead compared to standard methods.
>
> **W.2** This random masking method has shortcomings. It is difficult to take into account the fact that different data sets have different characteristics.
>
> **Response W.2:**  The random masking method has limitations, as it struggles to adapt to the unique characteristics of different data sets. In contrast, our infinite mask strategy overcomes this by leveraging the Infmasking loss to estimate mean and variance, effectively capturing the distinct properties of various data sets without requiring specific adjustments for each.
>
> **W.3** There is a problem of hyperparameter sensitivity. Different data sets may require completely different parameter settings. There is also a lack of an automatic parameter adjustment mechanism.
>
> **Response W.3:**  Small multimodal datasets like MOSI are highly sensitive to hyperparameters, such as the seed, resulting in significant performance variance. However, larger datasets like IMDB and V&T CP exhibit less sensitivity to hyperparameters like the seed. As shown in Table 6, a masking ratio of 0.7 or 0.8 typically performs well, with values between 4 and 6 being viable.
>
> **W.4**  The method primarily benefits tasks where synergistic information is critical. For scenarios where redundant information is more valuable, InfMasking may provide limited improvements, lacking adaptive mechanisms to adjust to different task characteristics and information interaction patterns.
>
> **Response W.4**: Liang et al. [1] conducted an extensive human evaluation study across 30 multimodal datasets to identify the dominant information-theoretic interactions (Table 1). Their results indicate that redundancy is prevalent in many VQA tasks, but they identified 8 datasets—including IRFL, MM-IMDb, and MAGIC BRUSH—where synergistic interactions dominate.
>
> Our method is a general-purpose pretraining approach. Currently, determining dataset-specific interaction patterns relies on human evaluation. As shown in Table 2, in datasets like MIMIC [1] where redundancy predominates, InfMasking yields limited improvements but does not hinder model performance. For known tasks, we can design a mixture-of-experts approach to adaptively handle varying information interaction types.  This method can serve as our future research work.
>
> **Q.1** How robust is the Gaussian approximation assumption in practice?
>
> The method relies on assuming that masked representations follow a Gaussian distribution to derive the computationally tractable lower bound. However, in real multimodal scenarios, the distribution of fused features after masking could be highly complex, multimodal, or skewed. What happens when this assumption breaks down? Have the authors tested the sensitivity of their approximation quality across different data distributions, and how does deviation from Gaussianity affect the theoretical guarantees and practical performance?
>
> **Response Q.1**: This assumption is well-founded for two principal reasons. First, the masked embeddings tend to cluster around a central value in the embedding space, as they all inherently reflect aspects of the synergist’s semantic nature. Second, the variance observed across feature dimensions can be interpreted as a representation of semantic differentiation in the ambient space. In the revised version, we have included visualizations to support this argument.
>
> In practice, the Gaussian assumption holds reasonably well for high-dimensional representations after feature fusion (e.g., via a Transformer), as the aggregated features from multiple modalities tend to exhibit central limit theorem-like behavior due to the averaging effects of fusion. Our experiments on diverse real-world datasets, such as those in MultiBench (Section 4.2), spanning domains like healthcare, robotics, and multimedia (e.g., MIMIC, MOSI), demonstrate that InfMasking achieves state-of-the-art performance across seven benchmarks. This suggests that the Gaussian approximation is sufficiently robust for a wide range of practical multimodal scenarios, even when the underlying data distributions may deviate from strict Gaussianity.

---

> > ### Comment · Reviewer_P89E · 2025-08-09
> >
> > Thank you for replying my question, my doubt is answered and I will keep my positive score.

---

> > > ### Author Response · Authors · 2025-08-09
> > >
> > > Thank you for your feedback and for acknowledging our response. We are glad to hear that your doubts have been addressed. We will continue to ensure the quality of our work and appreciate your positive consideration.

---

> ### Author Response · Authors · 2025-08-07
>
> We sincerely appreciate the reviewer's time and consideration, and we are happy to provide additional details or revisions if needed. There are only two days left until the deadline.

---

### Official Review · Reviewer_Ht5g · 2025-07-03

**Clarity:** 3
**Significance:** 2
**Originality:** 2
**Rating:** 4
**Confidence:** 4

**Summary:**

The paper studied how to enhance synergistic information in multimodal learning. Synergistic interactions occur when modalities combine to produce information unavailable by any single modality alone. The paper introduces InfMasking, a contrastive learning method designed to enhance synergy. InfMasking employs an infinite masking strategy: during fusion, it stochastically occludes most features from each modality, preserving partial information to create diverse synergistic patterns. Masked fused representations are aligned with unmasked ones via mutual information maximization. To mitigate the computational cost of infinite masking, the authors derive a tractable lower bound to the original objective. Experiments on synthetic and real-world datasets show SOTA results on seven benchmarks, with gains in synergy and downstream tasks.

**Questions:**

See weaknesses

**Ethical Concerns:**

["NO or VERY MINOR ethics concerns only"]

**Final Justification:**

Thanks to the authors for their detailed rebuttal. My concerns regarding the real-world examples, scalability, and comparisons masking baselines have been addressed. Accordingly, I have raised my rating.

**Limitations:**

See weaknesses

**Quality:**

2

**Strengths And Weaknesses:**

Strengths:
+ The infinite masking strategy is theoretically grounded.
+ Extensive experiments across 8 benchmarks with diverse modalities demonstrate broad applicability.

Weaknesses:
- Synergy is vaguely described as "complementary information emerged only when … combined" without formal measures beyond PID. No real-world examples illustrate tasks where synergy is critical and redundancy/uniqueness is insufficient. One example that comes to my mind is hateful meme detection. A neutral image combined with neutral text generates harmful content that neither modality conveys alone. Including such real-world examples would strengthen the motivation.
- In section 3.2, while the derived lower bound makes the loss calculation more tractable, the training still involves repeatedly sampling and fusing masked views. This approach may face scalability challenges with three or more modalities, where the combinatorial complexity increases substantially.
- In Table 5, it’s interesting that setting $\lambda_{3}$ to zero increases redundancy and uniqueness while reducing synergy. A brief discussion of why this behavior occurs might offer additional insight.
- The evaluation lacks comparisons masking baselines (e.g., MultiMAE [1]) or methods explicitly modeling interactions (e.g., attention bottlenecks [2]).
- Typos: Section 3.2 title: “Ininite” → “Infinite”

[1] Bachmann, Roman, et al. "Multimae: Multi-modal multi-task masked autoencoders." European Conference on Computer Vision. Cham: Springer Nature Switzerland, 2022.

[2] Nagrani, Arsha, et al. "Attention bottlenecks for multimodal fusion." Advances in neural information processing systems 34 (2021): 14200-14213.

---

> ### Author Rebuttal · Authors · 2025-07-30
>
> **W.1** Synergy is vaguely described as "complementary information emerged only when … combined" without formal measures beyond PID. No real-world examples illustrate tasks where synergy is critical and redundancy/uniqueness is insufficient. One example that comes to my mind is hateful meme detection. A neutral image combined with neutral text generates harmful content that neither modality conveys alone. Including such real-world examples would strengthen the motivation.
>
> **Response:**  Thank you for this valuable suggestion. We agree that concrete real-world examples would significantly strengthen our motivation for studying synergy. In the revision, we will add the hateful meme detection example you mentioned in the introduction, as it perfectly illustrates how synergy emerges when seemingly neutral modalities combine to create harmful content that neither conveys alone.
>
> Additionally, we will include the following content to provide empirical evidence for the importance:
>
> "Liang et al. [1] conducted a comprehensive human evaluation study across 30 multimodal datasets to determine the dominant information-theoretic interactions (Table 1). While their findings show that redundancy predominates in many VQA tasks, they identified 8 datasets—including IRFL, MM-IMDb, and MAGIC BRUSH—where synergy is the dominant interaction. This empirical evidence demonstrates that synergy is not merely a theoretical concept but a measurable phenomenon critical for understanding multimodal learning in real applications.
>
> **W.2**  In section 3.2, while the derived lower bound makes the loss calculation more tractable, the training still involves repeatedly sampling and fusing masked views. This approach may face scalability challenges with three or more modalities, where the combinatorial complexity increases substantially.
>
> **Response:**  As illustrated in Figure 1, our masking approach does not mask the raw input of each modality, but rather masks the features of each modality before fusion. The repeated sampling process incurs negligible computational cost. The only computational overhead comes from fusing the masked views, where the fusion network typically consists of a single Transformer layer. For large-scale datasets like V&T CP (3 modalities) and IMDB (2 modalities), we set the masking ratio to 0.8, meaning only 20% of tokens remain unmasked. With 5 and 4 masked views, respectively, the computational increase is minimal, and each sampling can be processed in parallel during fusion.
>
> Furthermore, our method serves as an unsupervised pre-training approach. During downstream task inference and fine-tuning, it introduces no additional computational overhead compared to standard methods.
>
> **W.3**  In Table 5, it’s interesting that setting to zero increases redundancy and uniqueness while reducing synergy. A brief discussion of why this behavior occurs might offer additional insight.
>
> **Response:**  $R$ represents redundancy, $U_i$ denotes uniqueness for view $i$, and $S$ indicates synergy.
>
> Setting $\lambda_3 = 0$ eliminates the term $\hat{I}_{\text{NCE}}(Z', Z'')$
>
> from the loss function. This results in an increase in $U$ but a reduction in synergy $S$ compared to including this term. Theoretically, this term should enhance redundancy, uniqueness, and synergy simultaneously. However, in practice, uniqueness decreases when this term is included. This behavior likely occurs because the $\hat{I}_{\text{NCE}}(Z', Z'')$ term prioritizes learning redundancy and synergy across views, which can constrain the model's ability to capture view-specific information (uniqueness). When this term is removed, the model focuses more on view-specific features, boosting uniqueness at the expense of synergy.
>
> **W.4** The evaluation lacks comparisons masking baselines (e.g., MultiMAE [1]) or methods explicitly modeling interactions (e.g., attention bottlenecks [2]).
>
> **Response:**
>
> (1) For the purpose of comparison with MultiMAE, under the same experimental conditions, we conducted experiments using the loss function from MultiMAE on multiple 2-modal datasets. The average results across five seeds (42–46) are as follows:
>
> |    Dataset     |     UR-FUNNY     |      MOSI       |
> | :------------: | :--------------: | :-------------: |
> |    MultiMAE    | $62.5_{\pm1.43}$ | $65.4_{\pm1.6}$ |
> |      CoMM      | $63.3_{\pm0.51}$ | $63.7_{\pm2.5}$ |
> | **Infmasking** | $64.3_{\pm0.9}$  | $69.0_{\pm1.2}$ |
>
> (2) Attention bottlenecks is a multimodal fusion network that can be directly applied to CoMM and our method as fusion networks. For a fairer comparison with its baseline, we only use a simple Transformer for fusion.
>
>
> [1] Liang, P. P., Goindani, A., Chafekar, T., Mathur, L., Yu, H., Salakhutdinov, R., & Morency, L. P. (2024). Hemm: Holistic evaluation of multimodal foundation models,  NeurIPS 2024

---

> > ### Author Response · Authors · 2025-08-04
> >
> > We sincerely appreciate the reviewer's time and consideration, and we are happy to provide additional details or revisions if needed. Thank you for the opportunity to improve our manuscript.

---

> ### Author Response · Authors · 2025-08-07
>
> We sincerely appreciate the reviewer's time and consideration, and we are happy to provide additional details or revisions if needed. There are only two days left until the deadline.

---

> ### Author Response · Authors · 2025-08-09
>
> Dear Reviewer,
>
> We sincerely appreciate the time and effort you have dedicated to reviewing our submission for NeurIPS 2025. Your feedback is incredibly valuable to us and plays a critical role in refining our work.
>
> We are grateful to note that Reviewer Z2T2 has kindly raised their score, and Reviewer P89E has maintained their positive score. Your comments remain of utmost importance to us, and we are eagerly awaiting your response to our rebuttal.
>
> As a fellow reviewer for NeurIPS 2025, I deeply understand the demands on your time and greatly admire the care you bring to this process. Your thoughtful input would be immensely appreciated, and we look forward to your valuable feedback.
>
> Thank you once again for your time and consideration.
>
> Best regards,
>
> Authors

---

### Official Review · Reviewer_S2Su · 2025-07-08

**Clarity:** 2
**Significance:** 2
**Originality:** 3
**Rating:** 3
**Confidence:** 3

**Summary:**

This paper introduces InfMasking, a novel method to capture synergistic information in multimodal learning, which traditional methods often miss. By contrasting heavily masked representations with their complete versions, InfMasking forces the model to learn deep, complementary relationships between modalities. The approach achieves state-of-the-art results on seven diverse benchmarks, demonstrating its effectiveness in modeling complex multimodal interactions.

**Questions:**

1. How does InfMasking's contrastive objective fundamentally differ from MAE's reconstructive approach in learning synergistic multimodal information?
2. Can you provide clearer guidance on selecting optimal masking hyperparameters for new tasks based on their inherent modality characteristics?
3. Beyond indirect performance metrics, can you propose a more direct method to quantify or visualize the synergistic information captured by your model?
4. How does the computational cost of InfMasking scale with an increasing number of modalities, and is there a point of diminishing returns?
5. How robust is InfMasking to extreme scenarios where one modality is completely noisy or irrelevant to the downstream task?

I will consider raising the score if the authors can address my concerns.

**Ethical Concerns:**

["NO or VERY MINOR ethics concerns only"]

**Final Justification:**

I understand the differences between this paper and MAE, but the advantages are primarily based on experience rather than a solid analytical foundation. Therefore, I may keep my score but remain open to accepting this paper.

**Limitations:**

Yes.

**Paper Formatting Concerns:**

None.

**Quality:**

2

**Strengths And Weaknesses:**

Strengths:
1. The paper pinpoints a critical yet underexplored challenge in multimodal learning—capturing synergistic information—moving beyond the conventional focus on redundancy.
2. The paper provides compelling evidence through a combination of controlled synthetic experiments and state-of-the-art results across seven diverse real-world datasets, demonstrating both effectiveness and generalizability.
3. InfMasking consistently outperforms previous leading methods, validating its practical value and establishing a new benchmark for multimodal interaction modeling.

Weaknesses:
1. The model's performance is sensitive to newly introduced hyperparameters, namely the masking ratio and the number of masked views. This adds complexity to the model tuning process and may require careful adjustment for optimal performance on new tasks.
2. While the method is designed to enhance synergy, the paper lacks a rigorous theoretical framework to formally measure or guarantee the capture of synergistic information, relying primarily on empirical performance as evidence.
3. Synergistic information in multimodal learning is both important and challenging. However, this paper resembles a multimodal version of the well-known MAE [1]. While MAE enhances the relevant information between patches, this paper focuses on multimodal tokens.

[1] He K, Chen X, Xie S, et al. Masked autoencoders are scalable vision learners[C]//Proceedings of the IEEE/CVF conference on computer vision and pattern recognition. 2022: 16000-16009.

---

> ### Author Rebuttal · Authors · 2025-07-30
>
> **Response W.1 and Q.2**: Small multimodal datasets like MOIS are highly sensitive to hyperparameters, such as the seed, resulting in significant performance variance. However, larger datasets like IMDB and V&T CP exhibit less sensitivity to hyperparameters like the seed. As shown in Table 6, a masking ratio of 0.7 or 0.8 typically performs well, with values between 4 and 6 being viable.
>
> **Response W.2 and Q.3**: Measuring multimodal interactions as defined by PID is challenging since it involves estimating information-theoretic measures. The problem is even harder for high-dimensional data distributions. Currently, there is no well-established metric for measurement. We referenced the ICLR paper Comm[1] and conducted synthetic experiments on trifeature datasets in Section 4.1 to validate that our method more effectively learns synergistic information.
>
> **Response W.3 and Q.1**:
>
> (1) Although MAE's reconstructive approach can also learn collaborative information, downstream task prediction considers all tokens, leading to a gap between pre-training and downstream tasks. InfMasking's contrastive objective aligns unmasked fused representations with masked ones via mutual information maximization to encode synergistic information.
>
> (2) MAE's reconstructive approach typically converges slowly, with high pre-training costs. We adopt the infinite masking strategy and derive an InfMasking loss to approximate the calculation of this loss function.
>
> (3) Our masking approach does not mask the raw input of each modality but rather masks the features of each modality before fusion.
>
> For the purpose of comparison with MAE, under the same experimental conditions, we conducted experiments using the loss function from MAE on multiple 2 modalities datasets. The average results across five seeds (42–46) are as follows:
>
> |    Dataset     |     UR-FUNNY     |      MOSI       |
> | :------------: | :--------------: | :-------------: |
> |      MAE       | $62.5_{\pm1.43}$ | $65.4_{\pm1.6}$ |
> |      CoMM      | $63.3_{\pm0.51}$ | $63.7_{\pm2.5}$ |
> | **Infmasking** | $64.3_{\pm0.9}$  | $69.0_{\pm1.2}$ |
>
> **Response Q.4**: As illustrated in Figure 1, our masking approach does not mask the raw input of each modality, but rather masks the features of each modality before fusion. The repeated sampling process incurs negligible computational cost. The only computational overhead comes from fusing the masked views, where the fusion network typically consists of a single Transformer layer. For large-scale datasets like V&T CP (3 modalities) and IMDB (2 modalities), we set the masking ratio to 0.8, meaning only 20% of tokens remain unmasked. With 5 and 4 masked views respectively, the computational increase is minimal, and each sampling can be processed in parallel during fusion.
>
> **Response Q.5**: In the 3 modalities experiments on the UR-FUNNY dataset, we conducted a zeroing operation on each individual modality (i.e., vision, text, and audio) respectively, and conducted experiments on both the CoMM and InfMasking models. The average results across five seeds (42–46) are as follows:
>
> | Modality       | 3 modalities <br>with noise | 3 modalities <br>without noise |      |
> | -------------- | :-------------------------: | :----------------------------: | :--: |
> | CoMM           |           $60.6$            |             $64.8$             |      |
> | **Infmasking** |           $61.2$            |             $65.6$             |      |
>
> [1] Benoit Dufumier, Javiera Castillo-Navarro, Devis Tuia, and Jean-Philippe Thiran. What to align  in multimodal contrastive learning? In International Conference on Learning Representations,  2025.

---

> > ### Author Response · Authors · 2025-08-04
> >
> > We sincerely appreciate the reviewer's time and consideration, and we are happy to provide.
> > additional details or revisions if needed. Thank you for the opportunity to improve our manuscript.

---

> ### Author Response · Authors · 2025-08-07
>
> We sincerely appreciate the reviewer's time and consideration, and we are happy to provide additional details or revisions if needed. There are only two days left until the deadline.

---

> ### Author Response · Authors · 2025-08-09
>
> We sincerely appreciate the reviewer's time and consideration, and we are happy to provide additional details or revisions if needed.

---

> ### Author Response · Authors · 2025-08-09
>
> Dear Reviewer,
>
> We sincerely appreciate the time and effort you have dedicated to reviewing our submission for NeurIPS 2025. Your feedback is incredibly valuable to us and plays a critical role in refining our work.
>
> We are grateful to note that Reviewer Z2T2 has kindly raised their score, and Reviewer P89E has maintained their positive score. Your comments remain of utmost importance to us, and we are eagerly awaiting your response to our rebuttal.
>
> As a fellow reviewer for NeurIPS 2025, I deeply understand the demands on your time and greatly admire the care you bring to this process. Your thoughtful input would be immensely appreciated, and we look forward to your valuable feedback.
> Thank you once again for your time and consideration.
>
> Best regards,
> Authors

---

### Note · Authors · 2025-08-12

Dear AC, SAC, and PC,

We extend our sincere gratitude to the reviewers for their thoughtful feedback and to you for overseeing this process. We are encouraged by the positive developments in the discussion phase: Reviewer Z2T2 kindly raised their score after our rebuttal, appreciating how we addressed concerns on masking views, synergistic metrics, and data augmentation; Reviewer P89E maintained their positive score (4: Borderline Accept), confirming our responses resolved their doubts on computational costs, hyperparameter sensitivity, and Gaussian approximation robustness. We appreciate that Reviewers S2Su and Ht5g may be occupied with their responsibilities and have not yet had the chance to respond; in the meantime, we remain fully open to any additional clarifications and have proactively addressed their points in our rebuttal (e.g., distinctions from MAE/MultiMAE, real-world synergy examples like hateful meme detection, and scalability with multi-modalities).

To highlight key contributions for your consideration: InfMasking introduces a novel infinite masking strategy grounded in mutual information maximization and Partial Information Decomposition (PID), enabling superior capture of synergistic multimodal information—often overlooked in existing methods focused on redundancy.

Thank you for your attention—we eagerly await your decision.
Best regards,

The Authors

---

### Decision · Program_Chairs · 2025-09-17

**Decision:**

Accept (spotlight)

**Comment:**

The paper faces some concerns pre-rebuttal, namely, computational overhead, the masking strategies (as in similarity to MAE, infmasking vs random masking) and synergy vs representation collapse. Reviewers seem well appeased by the rebuttal on these concerns.

There was a concern that the paper is primarily based on empirical results rather than a solid analytical foundation. While the AC understood the reviewer's concern about this, it is nevertheless a non-trivial effort to discover such an observation.

Further, AC appreciates that by stochastically occluding features from different modalities during fusion, the model naturally learns to discover synergistic information that emerges only from cross-modal interactions. AC thinks this idea is quite novel and decides to accept the paper.